

# Evolution of morphological and climatic adaptations in *Veronica L.* (Plantaginaceae)

Jian-Cheng Wang[1], Bo-Rong Pan[1] and Dirk C. Albach[2]

[1] Key Laboratory of Biogeography and Bioresource in Arid Land, Xinjiang Institute of Ecology and Geography, Chinese Academy of Sciences, Urumqi, PR China
[2] Institute for Biology and Environmental Sciences, Carl von Ossietzky-University Oldenburg, Oldenburg, Germany

## ABSTRACT

Perennials and annuals apply different strategies to adapt to the adverse environment, based on 'tolerance' and 'avoidance', respectively. To understand lifespan evolution and its impact on plant adaptability, we carried out a comparative study of perennials and annuals in the genus *Veronica* from a phylogenetic perspective. The results showed that ancestors of the genus *Veronica* were likely to be perennial plants. Annual life history of *Veronica* has evolved multiple times and subtrees with more annual species have a higher substitution rate. Annuals can adapt to more xeric habitats than perennials. This indicates that annuals are more drought-resistant than their perennial relatives. Due to adaptation to similar selective pressures, parallel evolution occurs in morphological characters among annual species of *Veronica*.

## INTRODUCTION

Flowering plants have repeatedly evolved a shorter life history of less than a year, with a record of less than three weeks from germination to seed set (*Cloudsley-Thompson & Chadwick, 1964*). The evolution of annual life cycles is combined with a monocarpic habit (i.e., death of the plant after first and only reproduction). Such plants are called annuals irrespective of considerable differences in their life histories (*Mortimer, Hance & Holly, 1990*) related to different ecology and habitats. The independent evolution of annuality in more than 100 different families from more than 30 orders of angiosperms (sensu *The Angiosperm Phylogeny Group, 2016*) and often even multiple times independently among closely related species (e.g., *Albach, Martinez-Ortega & Chase, 2004*; *Andreasen & Baldwin, 2001*; *Hellwig, 2004*; *Jakob, Meister & Blattner, 2004*; *Kadereit, 1984*) has made characterization of the annual habit difficult. Furthermore, the necessity to complete the life cycle within one season puts enormous constraints on plants that evolutionarily resulted in reduction in size to reach reproductive age faster and more reliably. Such a scenario has led to convergent evolution in several traits in annuals, especially a selfing breeding system but also a variety of other morphological, physiological, karyological and genomic traits (*Silvertown & Dodd, 1996*). This widespread convergence has given rise to misconceptions about the evolution

Corresponding author
Dirk C. Albach,
dirk.albach@uni-oldenburg.de

of annuals, particularly in cases when a rigorous phylogenetic hypothesis is lacking and comparative methods are not employed (*Albach, Martinez-Ortega & Chase, 2004*).

Several environmental factors that are not mutually exclusive can cause circumstances under which annuals have advantages over perennials, and most of these are related to the ability of annuals to survive unfavorable periods as seeds. Proposed factors include seasonal stress such as drought (*Macnair, 2007*; *Whyte, 1977*), heat (*Evans et al., 2005*), frost (*Tofts, 2004*; *Whyte, 1977*), unpredictable environment (*Stearns, 1976*), grazing/seed predation (*Klinkhamer, Kubo & Iwasa, 1997*; *Vesk, Leishman & Westoby, 2004*), flooding (*Kadereit, Mucina & Freitag, 2006*), limited maternal resources (*Hensel et al., 1994*), low competition (*Lacey, 1988*) and escape from pathogens over time (*Clay & Van der Putten, 1999*; *Thrall, Antonovics & Hall, 1993*). Even anthropogenic selection factors such as regular mowing and cultivation techniques may induce annual life history (*Baker, 1974*; *Hautekèete, Piquot & Van Dijk, 2002*). Therefore, it is often unclear whether evolutionary change is associated with annual life history per se or whether it is a reaction to a specific environmental condition. Advances in phylogeny reconstruction and comparative analyses allow investigation of the processes and the pattern of life history variation in more detail. Whereas a number of taxa have been analyzed in detail to infer the number of origins of annual life history and infer climatic circumstances of the shifts (e.g., *Datson, Murray & Steiner, 2008*; *Turini, Bräuchler & Heubl, 2010*) few employed rigorous comparative methods to analyze these shifts in life history. For example, *Drummond et al. (2012)* demonstrated increased speciation rates in derived montane perennial clades of *Lupinus* compared to lowland annuals. *Ogburn & Edwards (2015)* found perennials occupying cooler climatic niches than related annuals.

*Veronica* is a good model system to investigate this issue since annual life history has been shown to have evolved with convergent morphological characteristics multiple times in the same geographical region (*Albach, Martinez-Ortega & Chase, 2004*). *Veronica* comprises about 450 species and is the largest genus in the flowering plant family Plantaginaceae (*Albach & Meudt, 2010*). Most species—including all annuals—are distributed in the Northern Hemisphere but there is also an additional prominent radiation in the Australasian region (but without annuals). Life forms include herbaceous annuals or perennials, and also shrubs or small trees. About 10% of *Veronica* species are annuals, a life history which has originated at least six times independently in the genus (*Albach, Martinez-Ortega & Chase, 2004*). Chromosome numbers, phytochemistry and DNA sequence data support the polyphyly of annuals in the genus (*Albach & Chase, 2001*; *Müller & Albach, 2010*) However, despite the fact that many species of *Veronica* are widespread in accessible regions of the world, climate data has thus far not been included in any analysis of the genus. Also, morphological characters were mostly mapped on phylogenetic trees (e.g., *Albach, Martinez-Ortega & Chase, 2004*) but not included in a comparative analysis. Thus, crucial information to understand the evolution of the genus has, thus far, been excluded from analyses. In this study, we implemented a comparative analysis of morphological and climate data using phylogenetic methods to address the following two questions: (1) What convergent morphological trends are displayed in annuals? (2) Are there climatic factors that may favor annual life history? By answering these questions, we aim to expand

our understanding of the evolution of life history and its impact on the adaptability of plants. More specifically, we address the hypothesis that annual life history and selfing evolved in parallel in adaptation to drought. Therefore, we tested a correlation of life history with a number of characters, such as corolla diameter, known to be correlated with selfing in *Veronica* (*Scalone, Kolf & Albach, 2013*) and contrasted these with characters considered unrelated to mating system, such as leaf length. For environmental parameters, we specifically tested a number of bioclimatic parameters associated with precipitation and temperature to test the alternative hypothesis that annual life history is related to hot temperature. By including a range of morphological and climatological data, we want to infer more exactly, which characters are associated with the annual-selfing-syndrome.

## MATERIAL AND METHODS

A total of 81 individuals representing 81 species and all 12 subgenera of *Veronica*, were used to establish the phylogenetic tree in this study. Of these, sequences from 67 species were downloaded from GenBank from previous studies (*Albach & Meudt, 2010*), whereas sequences from 14 species, which were collected in Xinjiang Province of China, were newly generated for this study (see Table S1). Six individuals of five other genera of Veroniceae (*Lagotis*, *Picrorhiza*, *Wulfeniopsis*, *Wulfenia*, and *Veronicastrum*) were designated as outgroups. Genomic DNA extraction and purification was carried out using commercial kits according to manufacturer's instructions (D2485-02, OMEGA bio-tek). In accordance with the methods of *Albach & Meudt (2010)*, we carried out PCR, sequencing and phylogenetic tree reconstruction. DNA sequences of four regions were PCR-amplified, including nuclear ribosomal internal transcribed spacer region (ITS) with primers ITSA (*Blattner, 1999*) and ITS4 (*White et al., 1990*), plastid DNA (cpDNA) *trnL-trnL-trnF* with primers c and f (*Taberlet et al., 1991*), *rps16* with primers rpsF and rpsR2 (*Oxelman, Lidén & Berglund, 1997*), *psbA-trnH* with primers psbA (*Sang, Crawford & Stuessy, 1997*) and trnH (*Tate & Simpson, 2003*). A PCR program of 95 °C for 2 min, 36 cycles of: 95 °C for 1 min, 50–55 °C for 1 min, and 72 °C for 1.5–2 min, and finally 72 °C for 5 min and 10 °C hold, was used for all markers. DNA sequencing was performed by Sangon Biotech Co., Ltd (Shanghai, PR China). Bayesian inference methods were used to analyze the combined data set. Best fitting substitution models for the datasets were inferred using jModelTest 2.1.7 (*Darriba et al., 2012*). The Bayesian inference tree was built using MrBayes 3.2.5 (*Ronquist et al., 2012*) with the GTR+$\Gamma$ model using the Markov chain Monte Carlo (MCMC) for 1,000,000 generations with a burn-in of 250,000. The posterior probability (PP) was used to estimate nodal robustness. The stationarity of the runs was assessed using Tracer version 1.6 (*Rambaut et al., 2014*). We approximated divergence times using the function chronopl in the R package "ape" (*Paradis et al., 2015*).

We obtained morphological traits from field measurements and referenced from various flora, such as Flora of China (*Hong & Fischer, 1998*), Flora d'Italia (*Pignatti, 1982*), Flora of New Zealand (*Allan, 1961*), New Zealand Plant Conservation Network (http://nzpcn.org.nz/default.aspx). Plant traits were coded for each species according to characters and character states used by *Saeidi-Mehrvarz & Zarre (2004)*. In total 9 binary
characters about resource acquisition and reproductive characteristics were taken into consideration (character states and scoring matrix were shown in Tables S2 and S3).

We obtained GPS latitude/longitude data from the GBIF website (http://www.gbif.org/) for up to 500 occurrence records for each species using the function occ in the R package "spocc" (*Chamberlain, Ram & Hart, 2016*). Invalid, low accuracy or duplicate data were removed. GPS data of species collected by us were also added to the analysis. Bioclimatic variables were obtained for each of the geographical coordinates from WorldClim (www.worldclim.org) and processed using ArcGIS version 10.0. Climate data from each locality was acquired using the toolbox function "Extract Values to Points" and average values for each bioclimatic variable was calculated for each species. Drought and heat can affect annual and perennial relative fitness (*Macnair, 2007*; *Whyte, 1977*; *Evans et al., 2005*; *Pérez-Camacho et al., 2012*), and 7 related bioclimatic variables were selected for analysis (GBIF localities and corresponding climate data, average data were shown in Tables S4 and S5).

We used the function ace in the R package "ape" (*Paradis et al., 2015*) to estimate ancestral character states and the associated uncertainty for life history. Additionally, we also calculated phylogenetic signal using the function phylo.d in the package "caper" (*Orme et al., 2012*). The R package "iteRates" was used to implement the parametric rate comparison test and visualize areas on a tree undergoing differential substitution (*Fordyce, Shah & Fitzpatrick, 2014*). We have conducted phylogenetic comparative analysis. The function binaryPGLMM in the R package "ape" was used to perform comparative tests of morphological traits between annual and perennial plants. We tested climate data differences between annual and perennial plants using the function aov.phylo in the package "geiger" (*Harmon et al., 2008*).

## RESULTS

The phylogenetic relationships of *Veronica* from Bayesian inference of the four-marker dataset are shown in Fig. S1. The result of Bayesian phylogenetic analyses was assessed using Tracer with all ESSs >200 (after discarding a burn-in of 25%). The main clades of the phylogenetic tree were consistent with previous studies. The evolution and inferred ancestral life history in *Veronica* are shown in Fig. 1. Scaled likelihood of perennial life history at the root was 0.99. The *D* value as calculated in caper is a measure of phylogenetic signal in a binary trait, for which a value smaller than 0 indicates high correlation of the trait with phylogenetic differentiation and greater than 1 corresponds to a random or convergent pattern of evolution. The value of *D* for life history was $-0.55$, thus demonstrating relatively strong phylogenetic conservatism. This implies that lifespan is a relatively conservative trait and the change from perennial to annual, despite seven origins in the genus, is not a frequent occurrence.

Substitution rates (as measured by branch lengths) differ among clades within *Veronica* (Fig. 2). In general, clades with more annual species have faster substitution rates. The only significant increase in substitution rates subtends the clade of annual subgenera *Cochlidiosperma* and *Pellidosperma*, whereas most of the significant decreases in

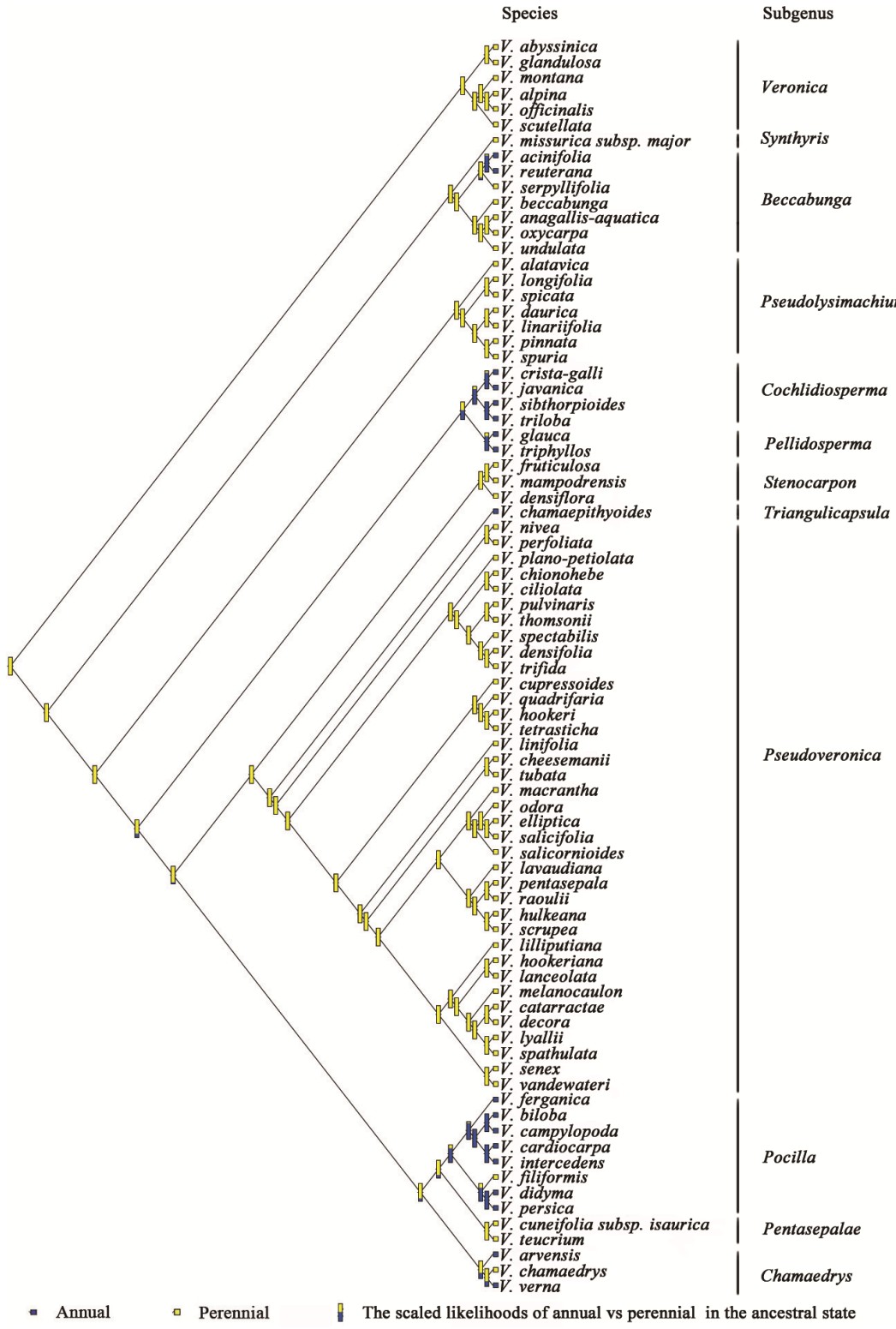

**Figure 1 Ancestral state reconstruction of life history in *Veronica*.** Proportional likelihoods for character states of ancestral life history are shown for nodes. Phylogenetic relationship of *Veronica* was built by Bayesian inference based on four-marker dataset. For Bayesian posterior probabilities, see Fig. S1.

**Table 1  Comparison of morphological traits between annual and perennial plants.**

| Mean rank | Annual | Perennial | $Z$ given phy | $P$ value given phy |
|---|---|---|---|---|
| Leaf length | 52.25 | 37.79 | −2.0971 | 0.03599 |
| Leaf width | 42.00 | 40.71 | −5e–04 | 0.9996 |
| Bract shape | 53.25 | 37.50 | −2.8681 | 0.004129 |
| Bract length | 31.50 | 43.71 | 2.2606 | 0.02378 |
| Corolla shape | 58.00 | 36.14 | −2.0321 | 0.04214 |
| Corolla diameter | 57.50 | 36.29 | −2.6443 | 0.008185 |
| Capsule apex | 60.50 | 35.43 | −2.7878 | 0.005307 |
| Stamen length | 51.75 | 37.93 | −2.1757 | 0.029581 |
| Style length | 42.50 | 40.57 | −0.0032 | 0.9975 |

**Table 2  Comparison of habitats between annual and perennial plants.** Temperature unit: (°C * 10); Precipitation unit:(mm).

| Variables | Annual | Perennial | $P$ value given phy |
|---|---|---|---|
| Max temperature of warmest month | 249.38 ± 8.22 | 199.59 ± 4.22 | 0.008 |
| Mean temperature of warmest quarter | 179.42 ± 6.78 | 138.05 ± 4.14 | 0.017 |
| Mean temperature of driest quarter | 90.90 ± 22.80 | 76.33 ± 9.19 | 0.741 |
| Annual precipitation | 695.94 ± 89.94 | 1539.67 ± 109.65 | 0.050 |
| Precipitation of driest month | 25.85 ± 4.26 | 80.24 ± 6.99 | 0.047 |
| Precipitation of driest quarter | 89.14 ± 13.49 | 287.47 ± 24.94 | 0.042 |
| Precipitation of warmest quarter | 168.35 ± 43.28 | 364.30 ± 27.03 | 0.092 |

substitution rates are associated with the evolution of the perennial, Australasian subgenus *Pseudoveronica*.

There are obvious differences in some morphological traits between annual and perennial plants (Table 1). Analysis of the morphological traits measured here shows that perennials have larger leaves, longer stamens and larger corollas, whereas annuals tend to have larger bracts and capsules with deeply emarginated apices.

Differences in habitats between annual and perennial plants are summarized in Table 2. The results demonstrated that annuals can withstand higher temperature (in warmest month). In terms of precipitation, there are also significant differences in precipitation of driest month. Perennials are found in areas of higher precipitation compared to annuals.

## DISCUSSION

The evolution of annual life history is a common evolutionary transition in angiosperms having occurred in more than 100 families. In angiosperms, the perennial habit is believed to be the ancestral condition (*Melzer et al., 2008*). Nevertheless, secondary evolution of perennial life history from annual herbaceous ancestors has been shown to occur in certain environments, such as islands (*Bohle, Hilger & Martin, 1996*; *Kim et al., 1996*) and mountains (*Karl et al., 2012*). Here, we analyzed a number of hypotheses regarding the evolution of annual life history in more detail based on comprehensive information on

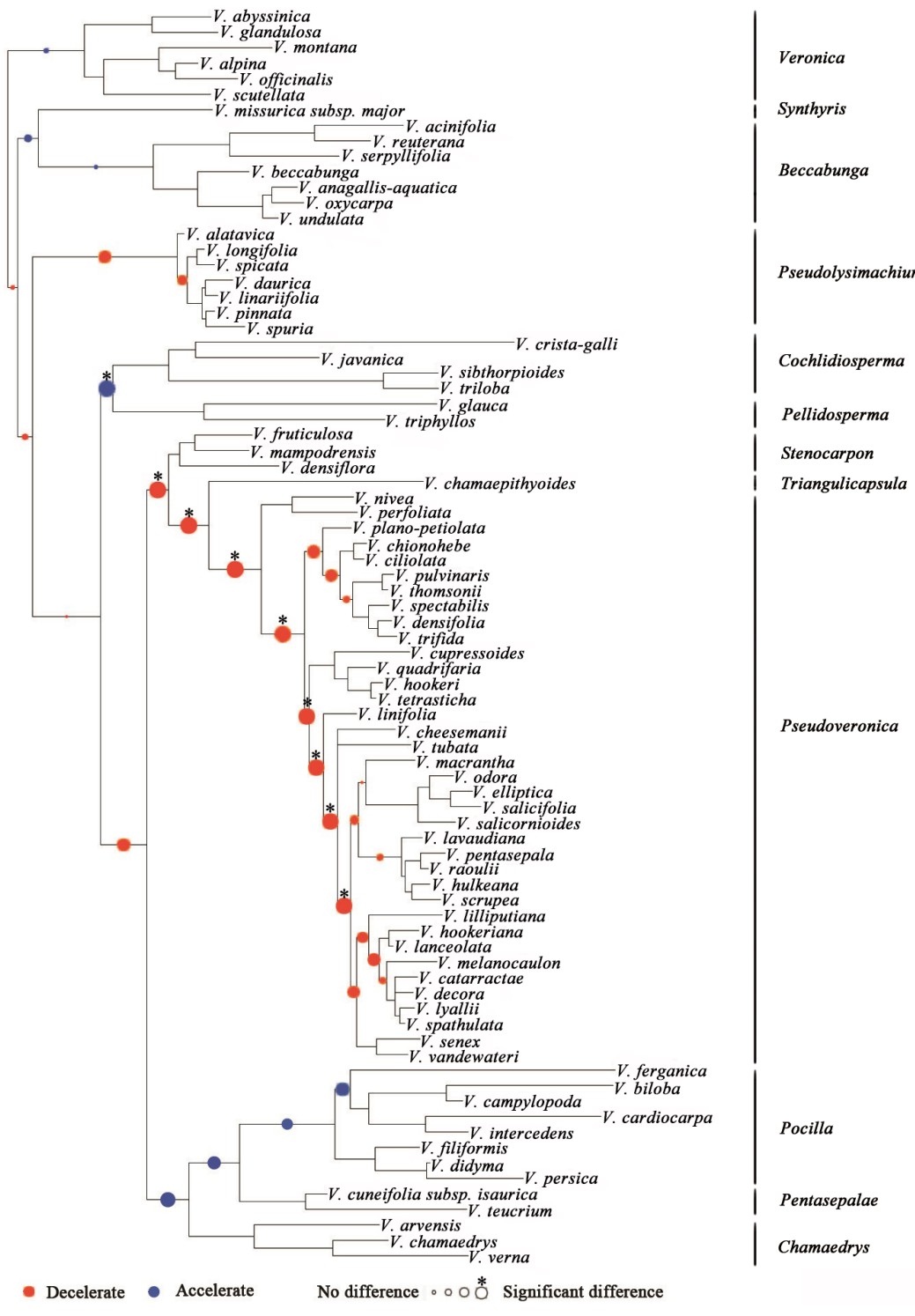

**Figure 2   Shifts in substitution rates in *Veronica* as assessed by the distribution of branch lengths among clades.** The blue nodes mean that substitution rates of that clade are faster than that of the remainder tree, whereas red nodes express the opposite meaning. The sizes of the colored nodes indicate the likelihood of rate-shifts. * The asterisk means that a rate-shift is significant. The results are based on limited sampling (<20%).

morphology and ecological data based on an explicit phylogenetic hypothesis. While many of these hypotheses were inferred in previous studies, modern comparative analytical tools allow to check these hypotheses in more detail. In this study, the ancestral condition of the genus *Veronica* has been inferred to have been perenniality and the annual life history has evolved multiple times with a single reversal in *V. filiformis* of the Caucasus Mountains consistent with previous conclusions (*Albach, Martinez-Ortega & Chase, 2004*). Overall, we inferred seven origins of annuals. An additional three origins of annuality (in *V. hispidula*, *V. peregrina* and *V. anagalloides* (all subgenus *Beccabunga*; *Albach, Martinez-Ortega & Chase, 2004*; *Müller & Albach, 2010*) are not included in the analysis here.

The seven to ten independent shifts between life histories are associated with considerable morphological diversity among annual species. However, certain characters are characteristic for annuals (the annuality syndrome) associated with the rapid completion of the life cycle. For example, the generation-time hypothesis, which assumes that mutations are mostly accumulated during recombination, states that organisms that reproduce faster such as annuals also have more DNA substitutions over time (*Page & Holmes, 2009*). Results of this study demonstrate that clades including annuals have a higher substitution rate and are, thus, consistent with this theory and previous analyses for *Veronica* (*Müller & Albach, 2010*), although this is significant only for the oldest clade of annuals (*V*. subg. *Cochlidiosperma* (Rchb.) M. M. Mart. Ort. & Albach). On the other side, the perennial clade with the lowest substitution rate (*V*. subg. *Pseudoveronica*, see above) is also the one with the highest diversification rate (*Meudt et al., 2015*). However, the impact of life history transformation is not restricted to substitution rate.

Two of the correlations detected are most likely associated with the smaller stature of annuals. These are the larger leaves of perennials and the larger bracts in annuals (especially in subgenera *Pocilla* and *Cochlidiosperma*) that compensate for the reduced number and size of stem leaves in smaller plants. Also, reduction to a single, terminal inflorescence is likely to be a consequence of small size but may also be related to differences in breeding system. Other inflorescence characters are more clearly associated with differences in breeding system between annuals and perennials.

Estimates for selfing among angiosperms as a whole are 25–30% (*Barrett & Eckert, 1990*) with estimates for annuals alone going up to 50% (*Hamrick & Godt, 1996*). The association between annual life history and selfing has been known for some time (*Henslow, 1879*) and has also been thoroughly discussed in the literature (e.g., *Barrett, Harder & Worley, 1996*; *Stebbins, 1957*). Annual species invest fewer resources into their sexual organs (e.g., number of lateral inflorescences; density of inflorescence, corolla size) than perennials (although not necessarily relative to overall size of the plants). Such changes are likely to be associated with parallel changes in life history and breeding system. A larger corolla and longer stamens have previously been demonstrated to be correlated with an outcrossing breeding system in the genus (*Scalone, Kolf & Albach, 2013*). Surprisingly, a longer style is here not associated with perenniality as inferred by *Scalone, Kolf & Albach (2013)*. In contrast, we infer that selfing is facilitated by lowering the stigma below the anthers through emargination of the capsule. By that means, the stigma is removed from the anthers without shortening the style. Other characters that may have an influence on breeding system in

perennials is the trend towards tubular corollas, which may contain more nectar, and the longer pedicels in perennials that allows better presentation of the flower. Thus, our analysis supports the notion that outcrossing is associated with perennial life history in *Veronica* (*Albach & Greilhuber, 2004*). Such a correlation in the evolution of annual life history is often argued to be due to reproductive assurance in annuals, depending on reproduction in their single season of flowering (*Busch & Delph, 2012*). However, to understand the basis for this association, one needs to move beyond such correlation and understand the ecological circumstances of transitions in life history.

Several such circumstances have been inferred to be responsible for the evolution of annual life history (see 'Introduction'). Here, we inferred higher temperature, higher temperature variation and lower precipitation to be the characteristic environmental conditions for annuals in comparison with perennials. This is consistent with previous suggestions that inferred drought, heat or unpredictable environment are responsible for the evolution of annual life history (*Evans et al., 2005*; *Stearns, 1976*; *Whyte, 1977*). Thus, despite the multiple origins of annuals in the genus, annual clades in *Veronica* may have reacted to the same climatic circumstances favoring a change in life history. Although we did not specifically test for differences among clades of annuals, markedly different climatic circumstances in one clade of annuals should have led to differences between inferences based on phylogenetically informed and non-phylogenetic analyses.

Consequently, it is likely that parallel evolution in different groups of *Veronica* led to the evolution of annual life history and a characteristic set of related characters. Parallel evolution is more likely if occurring in the same region at the same time because of the same selection pressure. Based on the molecular dating of *Veronica* in *Meudt et al. (2015)*, however, annual lineages originated over a range of dates starting in the Miocene, similar to other Mediterranean annuals inferred to have originated in response to the evolution of the Mediterranean climate evolution and the Messinian salinity crisis (*Fiz, Valcárcel & Vargas, 2002*). With the exception of *V. peregrina*, not included here, all groups of annual *Veronica* originated from ancestors in the Mediterranean and southwest Asia. Thus, progressing aridification may have spurred evolution of annual life history at different times in the same region in different groups of *Veronica*. During aridification, competition from related species decreased, and environmental filtering became a major limiting effect on species. Under such circumstances, the avoidance strategy of annuals by drought-tolerant seeds is favored by natural selection (*De Bello, LepŠ & Sebastia, 2005*). However, this hypothesis will be investigated in more detail in the different clades of annual *Veronica* by more detailed investigation of character evolution and ancestral habitat estimation.

### Funding

This work was supported by the West Light Talents Cultivation Program of Chinese Academy of Sciences (XBBS201202) and the National Natural Science Foundation of China (Grant No. 31400208). The funders had no role in study design, data collection and analysis, decision to publish, or preparation of the manuscript.

## Grant Disclosures

The following grant information was disclosed by the authors:
West Light Talents Cultivation Program of Chinese Academy of Sciences: XBBS201202.
National Natural Science Foundation of China: 31400208.

## Competing Interests

The authors declare there are no competing interests.

## Author Contributions

- Jian-Cheng Wang conceived and designed the experiments, performed the experiments, analyzed the data, wrote the paper, prepared figures and/or tables, and collected 14 species distributed in Xinjiang Province of China.
- Bo-Rong collected 14 species distributed in Xinjiang Province of China.
- Dirk C. Albach contributed reagents/materials/analysis tools, wrote the paper, reviewed drafts of the paper.

## DNA Deposition

The following information was supplied regarding the deposition of DNA sequences:

ITS trnL-trnL-trnF rps16 psbA-trnH

AF313028 AF486416 AY218805 FJ848085
AF509814 AF486415 FJ848212 FJ848086
AF509813 AF486414 AY218806 FJ848090
AF313030 AF486412 AY218802 FJ848087
AF313025 AF486409 AY218804 FJ848088
FJ848064 AF486411 AY218808 FJ848089
AF313009 AF513350 FJ848213 FJ848092
AF509798 AF486399 FJ848215 FJ848098
KU047989 KU048030 KU048016 KU048003
AF313013 AF486387 FJ848214 FJ848093
KU047995 KU048036 KU048022 KU048009
AF313002 AF486380 FJ848224 FJ848115
AF313015 AF486403 AY218820 FJ848099
KU047992 KU048033 KU048019 KU048006
AF486364 AY673624 AY218811 FJ848117
KU047990 KU048031 KU048017 KU048004
AY034859 AY540887,AY540901 FJ848253 FJ848148
AF313003 AF486377 AY218814 FJ848116
AF509796 AF511477/AF511478 FJ848222 FJ848112
FJ848078 FJ848053 FJ848254 FJ848149
FJ848070 FJ848044 FJ848233 FJ848127
FJ848067 FJ848041 FJ848230 FJ848124
AF509799 AF486367 AY218816 FJ848109
AF486354 AF486372 AY218804 FJ848121
AF037378 AY540880,AY540894 FJ848251 FJ848145

AF313023 AF511479/AF511480 FJ848217 FJ848102
AF229047 AY540877 FJ848255 FJ848150
FJ848068 FJ848042 FJ848231 FJ848125
AF509818 AF486369 FJ848227 FJ848120
AF037393 AY540883,AY540897 FJ848242 FJ848138
KU047984 KU048025 KU242588 KU047998
AF486363 AF486368 FJ848225 FJ848118
AF313004 AF486383 AY218812 FJ848113
AF313008 AF486394 AY218822 FJ848094
AF313006 AF486395 FJ848220 FJ848108
FJ848077 FJ848050 FJ848250 FJ848144
FJ848079 FJ848054 FJ848256 FJ848151
FJ848075 n.a. FJ848238 FJ848132
AY673609 AY673628 FJ848226 FJ848119
AY540867 AY540872 FJ848219 FJ848106
n.a. FJ848055 FJ848257 FJ848152
AF229043 n.a. FJ848239 FJ848133
AF037394 FJ848052 AY21813 FJ848147
KU047993 KU048034 KU048020 KU048007
FJ848080 FJ848056 FJ848258 FJ848153
AF313021 AF486407 AY218818 FJ848103
FJ848081 FJ848057 FJ848259 FJ848154
AY034853 FJ848048 FJ848244 FJ848137
DQ227331 DQ227337 FJ848223 FJ848114
AF037396 FJ848058 FJ848260 FJ848155
AF313019 AF486397 AY218819 FJ848105
AF313014 AF486388 AY218824 FJ848095
AF037382 AY540890,AY540903 FJ848229 FJ848123
AF037388 AY540882,AY540896 FJ848245 FJ848139
AF313012 AF486391 n.a. FJ848096
KU047987 KU048028 KU048014 KU048001
FJ848076 n.a. FJ848240 FJ848134
FJ848066 FJ848040 FJ848228 FJ848122
KU047996 KU048037 KU048023 KU048010
KU047997 KU048038 KU048024 KU048011
AF229050 FJ848059 FJ848261 FJ848156
FJ848072 FJ848046 FJ848235 FJ848129
AF037377 AY540886,AY540900 FJ848249 FJ848143
AF037380 AY540885,AY540899 FJ848241 FJ848135
AY540866 AY486447 FJ848216 FJ848100
AF037386 FJ848049 FJ848248 FJ848142
AF069465 AY540879,AY540893 FJ848246 FJ848140
FJ848074 n.a. FJ848237 FJ848131
AF509805 AF486393 AY218823 FJ848097
FJ848082 FJ848060 FJ848262 FJ848157
AF313017 AF486400 AY218821 FJ848101
AY850099 AY540876 FJ848221 FJ848110
KU047985 KU048026 KU048012 KU047999
AF229051 FJ848061 FJ848263 FJ848158
AF229044 FJ848062 FJ848264 FJ848159
AF313022 AF486405 FJ848218 FJ848104
KU047988 KU048029 KU048015 KU048002
AY034866,AY034867 FJ848051 FJ848252 FJ848146
KU047986 KU048027 KU048013 KU048000
FJ848073 FJ848047 FJ848236 FJ848130
FJ848083 FJ848063 FJ848265 FJ848160
AF509804 AF513333 AY218815 FJ848111
FJ848065 FJ848039 AY218817 FJ848107
AY540870 AY540874 n.a. n.a.
KU047994 KU048035 KU048021 KU048008
FJ848084 AF486381 AY21813 FJ848161
KU047991 KU048032 KU048018 KU048005.

## Data Availability

All the data acquired from the GPIF website (http://www.gbif.org/) and the GPS data collected by the authors is contained in Table S4. Table S5 is a summary table with the average data used for each of the 81 species.

## Supplemental Information

Supplemental information for this article can be found online at http://dx.doi.org/10.7717/peerj.2333#supplemental-information.

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
