# Peer review of "Evolution of morphological and climatic adaptations in Veronica L. (Plantaginaceae)"

_PeerJ, doi:10.7717/peerj.2333_

## Round 0.1 · original submission · Major Revisions

The reviewers provide numerous constructive comments and I would encourage the authors to consider them carefully. Please provide a point by point account of how you address these comments and, should you disagree with the reviewers on certain points, explain clearly why. I look forward to see a revised version of this manuscript.

Reviewer 1 ·

Basic reporting

The standard of English language is generally good, however having a native speaker edit the revised manuscript before publication would be advisable.

I am confused about the title of this paper. Is it: “Evolution of morphological and climatic adaptations in Veronica L. (Plantaginaceae)” or “Evolution of life history and its morphological and climatic correlates in Veronica (Plantaginaceae”?

Citations to appropriate literature should be added for the following parts of the manuscript:
-“Such a scenario has led to convergent evolution in several traits in annuals, especially a selfing breeding system but also a variety of other morphological, physiological, karyological and genomic traits.” (Introduction)
-“This widespread convergence has given rise to misconceptions about the evolution of annuals, particularly when in cases when a rigorous phylogenetic hypothesis is lacking and comparative methods are not employed.” (Introduction)
-The GenBank sequences of 4 markers x 67 species used in this study (cite original papers which produced the original sequences). (Materials & Methods)
-“We obtained species morphological traits from field measurements and referenced from various flora, such as Flora of China (Hong & Fischer, 1998), Flora d´Italia (Pignatti, 1982).” At least reference for the Southern Hemisphere species is required here, but I would prefer ALL references cited here, not just two examples (Materials & Methods)

Figures are relevant, although they are hard to see when reduced in size in the peer review pdf document, but look ok as their own pdfs. I have suggested some minor changes and additions to the figures and the figure captions for improving clarity and fixing some typos.

Experimental design

Some important info is missing from Table 1. Voucher specimens (collector, collection number, and herbarium where located) should be added to Table 1 for the 14 individuals that were newly sequenced in this study. At the moment only undecipherable info such as “”xy015-2” or “W26-3” are included in the “voucher” column of Table 1 for these individuals. In addition, a column showing whether plants are perennial or annual should be added to this table.

A new table should be added with the scored morphological data matrix.

The last sentence of Mueller & Albach 2010 suggests a research gap with this paper aims to partially fill. Their sentence and that reference should be explicitly cited in the Introduction.

Comments regarding the GBIF data and bioclimactic analyses. I see this as potentially the most problematic aspect of the paper:
-The information provided regarding the GBIF localities is not specific enough to replicate.
- When I search for these species, I get different numbers of localities used in the paper based on the 81 Excel spreadsheet vs. what can be found on GBIF. So, what filters were used on the GBIF website for downloading up to 500 localities per species? (For example, were only “specimens” used (this would be preferred) or were other data types used? I would not necessarily trust identifications for all observations from iNaturalist.org for example.) If there were more than 500 localities found on the GBIF website, what were the criteria used for selecting 500?
-What are the 13 columns between “alt” and the 19 bioclimatic variables in the 81 excel files (i.e. sq1, s2…sc2)? Are these used in any way in the paper and analyses?
-The authors should furthermore provide two new supplementary data files: 1) A new supplementary table with ALL GBIF localities and corresponding climate data together with the herbarium voucher information for that locality (which is also provided by GBIF), i.e. create one large Excel file by merging all 81 Excel climate files provided with the submission into one large Excel file, but remove the summary statistics and add the voucher info so each specimen can be traced, and 2) Another new supplementary table with one line per species, i.e. the AVERAGE data used for each of the 81 species.

Validity of the findings

With respect to the phylogenetic analyses, although the authors include 14 newly sequenced individuals, I am not sure of significance of adding these specific individuals, as they are not discussed anywhere in the manuscript, and they do not seem to alter in any way previous Veronica topologies.

Testing whether the averages of bioclimactic variables of all perennial vs. annual species (which are themselves based on averages of bioclimatic variables over multiple data points for each individual species) seems to be an extremely blunt tool to test whether bioclimactic variables correlate with life history. There is so much data summarised (averages of averages) and lost in such an analysis, that I doubt whether the results are truly meaningful.

How do the results from this study go beyond (rather than just confirm) what has already been reported in previous Veronica studies on phylogeny and life history? Possibly the information is in the discussion, but I feel this information needs to be more explicitly stated or highlighted.

Additional comments

This paper uses phylogenetic analyses of sequences from 4 DNA markers from 67 species downloaded from GenBank and another 14 species collected in China newly sequenced here. It also uses 19 bioclimatic variables averaged over multiple localities (downloaded from GBIF) for each species, as well as morphological data from the literature (again averaged for each species) to study the evolution of life history and its morphological and climatic correlates in Veronica (Plantaginaceae). The main aims of the paper are to assess convergent morphological trends and climatic factors in the genus, and relate these to the evolution of life history (annual vs. perennial).

The paper uses standard molecular phylogenetics (of 1 individual per species for 81 species) to investigate life history evolution in Veronica, building on previous studies by the last author and colleagues (Albach et al. 2004; Mueller & Albach 2010; Albach & Meudt 2010; Meudt et al. 2015). My main concern is whether or not the methods, sampling and analyses in this paper actually address the question, and whether the paper goes beyond what these previous studies have already shown to address a knowledge gap. The most novel aspect of the paper is introducing bioclimactic data to the analyses, which could help address an important research gap (Mueller & Albach 2010). However, both the morphological and climactic analyses as done here are rather coarse tests, using representative (average) data for each species, and then grouping these species again into two groups (annuals vs perennials) to look for significant differences between the averages of these groups (with and without taking into account the phylogeny). Perhaps this paper, and the analyses in it, might be acceptable as a “first pass” preliminary study, but this needs to be explicitly mentioned/discussed in the manuscript and title to this effect. What the authors suggest at the end of the paper, i.e. “However, this hypothesis will be investigated in more detail in the different clades of annual Veronica by more detailed investigation of character evolution and ancestral habitat estimation”, sounds like it could be a much better way to test the questions, especially if it involves much more sampling within species for morphological, climactic and DNA sequence data.

I have annotated the manuscript Word document with the above comments and other additional comments and edits. I hope they are useful to the authors for improving the paper.

Annotated reviews are not available for download in order to protect the identity of reviewers who chose to remain anonymous.

·

Basic reporting

Nowhere on the tree are we given node posterior probabilities, so there is no way to assess support. This absolutely needs to be included – I am always suspicious of trees for which no pp’s or support values are given, even in supplementary info. It would be preferable for these to be included in the main body of the paper.

Table 1 is probably not necessary to include in the main text of the article and may be best as a supplementary table.

Ideally, figure 1 and 2 should be combined in some way so that we can more easily and quickly assess the relationship between life history shifts and rate changes. It’s a bit unclear what the three boxes on each node in figure 1 are showing us. I can figure out that the number of boxes of a given color is supposed to represent the likelihood for a given state at that node, but this should be spelled out in the figure caption. Also, why not just use pie charts? There are only two states, so they would be fairly unambiguous.

To me, table 3 would be easier to understand if the first column had the variable descriptions, even if abbreviated.

I don’t understand the sentence starting on line 177: “In contrast, we infer that selfing is facilitated by lowering the stigma below the anthers through emargination of the capsule.” I understand the capsule to refer to a dry, dehiscent fruit, so how does that affect the spatial relationship of the stigma and anthers?

The authors should cite Ogburn and Edwards 2015, which takes a similar approach to look at similar questions. Other citations pertaining to life history and climate niche evolution that were not included and probably should be include Drummond et al 2012, Sun et al. 2012, Tank and Olmstead 2008, and Datson et al. 2008.

Experimental design

I wrote my review as one piece - please see "General comments to authors" for my feedback on experimental design and validity of findings.

Validity of the findings

I wrote my review as one piece - please see "General comments to authors" for my feedback on experimental design and validity of findings.

Additional comments

The authors performed a comparative analysis of life history evolution in the genus Veronica, which has independently evolved the annual habit from perennials multiple times (at least seven times according to the authors). The examined relationships between life history state and a suite of morphological traits, as well as with the standard 19 BIOCLIM variables + elevation. The results are fairly interesting, especially regarding the analysis of convergence in some of the floral traits in annual Veronica, but unfortunately the study suffers from some issues around tree inference and especially around running multiple statistical tests without considering the possibility of type I errors.

First, but more easily fixable, the tree inference is a little thin. Partitioning of genetic markers and choice of evolutionary models can have a large effect on tree inference. Unfortunately, there is no indication that any of these important pre-analysis steps were taken. Were the genetic markers partitioned or not? How was this decision made? How was the GTR+gamma model chosen? In short, was any kind of model testing software run? Please do so if it was not done for the initial analysis.

A single run of 1,000,000 MCMC generations seems very minimal for a Bayesian tree inference analysis. If the authors used the default of 2 simultaneous runs in MrBayes, that helps a bit, although we are not told if this was the case. Four runs from independently derived random start trees is to me a minimum, so the analysis should be run twice to give four independent runs total. Also, the analysis should probably go for a longer number of generations – I would think at least 10,000,000 in my experience. If the authors do feel that 1,000,000 generations is sufficient, they should indicate how stationarity of the MCMC chains for various parameters was assessed and provide evidence of such.

GBIF records are used for the spatial aspect of the analysis. The authors mention an upper limit of 500 records per species, but don’t give a rationale for that. I would think that the bigger concern would be species with only a handful of records. I checked the supplemental files provided and many (Veronica alatavica, V. daurica, V. densiflora, V. spectabilis, V. vandewateri) consist of a single record, while at least one other (V. cardiocarpa) consists of two. I leave it to the authors to decide what a minimum acceptable number of records is, but regardless this information should be more readily available to the reader. It is often possible to find more records from other databases (frequently better curated) than GBIF. Additionally, as a massive database, much of the information in GBIF is of fairly low quality – were any steps taken to “clean” the data – i.e., remove records that were clearly in the wrong place, etc? Please indicate the steps taken in the manuscript.

With the analysis of molecular rate variation in annuals and perennials, I would be very cautious about interpreting the meaning of substitution rate variation when less than 20% of species in the clade have been sampled. I don’t think the authors should not do this analysis, because the general picture of rate variation as a function of life history is interesting and probably true, but they should acknowledge some caveats about limited sampling here.

The end of the introduction provides us with no hypothesis, we are simply told the authors are asking general questions about “1) what convergent trends are displayed on morphological traits in annuals? 2) Are there climatic factors that may repeatedly favor annual life history?” This makes it sound as if the entire study is a fishing expedition – I don’t believe the authors approached their study this way, but they should set up their introduction with clearer hypotheses. For example, if the authors expected certain floral traits to correlate with the annual habit, they should explain what those traits are and what evidence or rationale there is for this expectation. Likewise with climate variables. This is highly preferable to saying “we are going to test a bunch of variables and see which ones show a relationship”, which is basically the way it is presented here.

By the same token, in the results we can see that they did statistical tests against many variables - 22 morphological variables and 20 climatic variables. With this many independent tests there is a high probability of type I errors – in fact you are basically guaranteed at least one – and the appropriate thing to do in this case is to apply a Bonferroni correction to their p-value threshold. If the authors had planned this ahead of time, they might have been more selective about which variables to test, but this is something that has to be done a priori.

I also disagree with using “traditional” aphylogenetic correlation methods side-by-side and basically interchangeable with phylogenetically explicit methods, especially when the authors have already shown us that there is strong phylogenetic signal for life history. If there is strong phylogenetic signal, then the distribution of the trait depends on the structure of the tree. While the results of both types of analysis generally agree, the phylogenetic methods typically have higher p-values than the standard correlations – in these cases, the significance levels of the aphylogenetic methods are happily adopted in the discussion without a mention. This is especially the case for the BIOCLIM variables, of which only two were significant at an alpha of 0.05 in the phylogenetic analysis. Statistically, this is a tough situation, especially when your predictor variable is categorical – you are reduced from 81 independent points to seven in the phylogenetically explicit analysis. This is of course not helped by having to apply the Bonferroni corrections, but statistically it is the right thing to do. I think the best thing for the authors to do is simply acknowledge that they have low statistical power to detect effects and talk about trends – future studies can try to follow up on these trends by adding to the tree with other taxa that have independently evolved an annual habit (mentioned in the discussion) to increase statistical power, and being more targeted about which variables to test. Other possible ways to analyze the data for now include using an independent-contrasts analysis like CAIC from the R package ‘caper’. This gives you contrasts, which you can analyze with a t-test to see if they are significantly different from 0. Another possibility is doing a linear model in which the lambda measure of phylogenetic signal is estimated on the model residuals – if signal in the linear model is low, the model will be adjusted more toward an aphylogenetic model, if signal is high, the model will be closer a strict Brownian motion model of trait evolution. Information about this here: http://blog.phytools.org/2012/11/fitting-model-in-phylogenetic.html and in Revell 2010. The catch is I’m not sure this will work with a categorical predictor variable. If the authors feel strongly that they want to keep the aphylogenetic analyses in the paper, they should clearly elaborate why this is justified, both to the reviewers and in the manuscript.

---

## Round 0.2 · accepted · Accept

Thank you for providing a point by point account of the changes you performed in the revised version of your manuscript. I think that the changes you made greatly improved your manuscript and I am happy to recommend publication in PeerJ.